# CLIPORT: What and Where Pathways
# for Robotic Manipulation

**Mohit Shridhar** [1,†]  **Lucas Manuelli** [2]  **Dieter Fox** [1,2]
[1]University of Washington    [2]NVIDIA
mshr@cs.washington.edu   lmanuelli@nvidia.com   fox@cs.washington.edu

[cliport.github.io](cliport.github.io)

**Abstract:** How can we imbue robots with the ability to manipulate objects precisely but also to reason about them in terms of abstract concepts? Recent works in manipulation have shown that end-to-end networks can learn dexterous skills that require precise spatial reasoning, but these methods often fail to generalize to new goals or quickly learn transferable concepts across tasks. In parallel, there has been great progress in learning generalizable semantic representations for vision and language by training on large-scale internet data, however these representations lack the spatial understanding necessary for fine-grained manipulation. To this end, we propose a framework that combines the best of both worlds: a two-stream architecture with semantic and spatial pathways for vision-based manipulation. Specifically, we present CLIPORT, a language-conditioned imitation-learning agent that combines the broad semantic understanding (*what*) of CLIP [1] with the spatial precision (*where*) of Transporter [2]. Our end-to-end framework is capable of solving a variety of language-specified tabletop tasks from packing unseen objects to folding cloths, all without any explicit representations of object poses, instance segmentations, memory, symbolic states, or syntactic structures. Experiments in simulated and real-world settings show that our approach is data efficient in few-shot settings and generalizes effectively to seen and unseen semantic concepts. We even learn one multi-task policy for 10 simulated and 9 real-world tasks that is better or comparable to single-task policies.

## 1 Introduction

Ask a person to "get a scoop of coffee beans" or "fold the cloth in half" and they can naturally take concepts like *scoop* or *fold* and ground them in concrete physical actions within an accuracy of a few centimeters. We humans do this intuitively, without explicit geometric or kinematic models of coffee beans or cloths. Moreover, we can generalize to a broad range of tasks and concepts from a minimal set of examples on what needs to be achieved. How can we imbue robots with this ability to efficiently ground abstract semantic concepts in precise spatial reasoning?

Recently, a number of end-to-end frameworks have been proposed for vision-based manipulation [2, 3, 4, 5]. While these methods do not use any explicit representations of object poses, instance segmentations, or symbolic states, they can only replicate demonstrations with a narrow range of variability and have no notion of the semantics underlying the tasks. Switching from packing red pens to blue pens involves collecting a new training set [2], or if using goal-conditioned policies, involves the user providing a goal-image from the scene [5, 6]. In realistic human-robot interaction settings, collecting additional demonstrations or providing goal-images is often infeasible and unscalable. A natural solution to both these problems is to condition policies with natural language. Language provides an intuitive interface for specifying goals and also for implicitly transferring concepts across tasks. While language-grounding for manipulation has been explored in the past [7, 8, 9, 10], these pipelines are limited by object-centric representations that cannot handle granular or deformable objects and often do not reason about perception and action in an integrated manner. In parallel, there has been great progress in learning models for visual representations [11, 12] and aligning representations of vision and language [13, 14, 15] by training on large-scale internet data. However, these models lack a fine-grained understanding on how to manipulate objects, i.e. physical affordances.

---

[†]Work done partly while the author was a part-time intern at NVIDIA.

5th Conference on Robot Learning (CoRL 2021), London, UK.

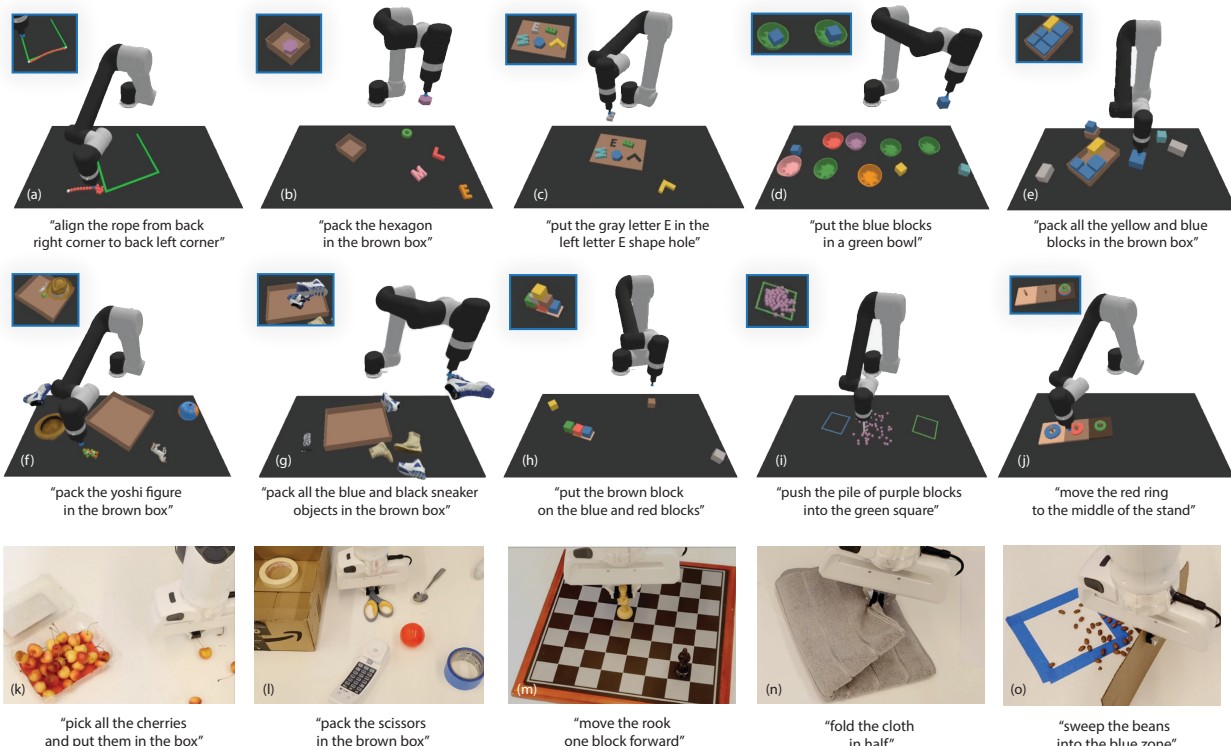

Figure 1. **Language-Conditioned Manipulation Tasks:** CLIPORT is a broad framework applicable to a wide range of language-conditioned manipulation tasks in tabletop settings. We conduct large-scale experiments in Ravens [2] on 10 simulated tasks (a-j) with 1000s of unique instances per task. See Appendix A for challenges pertaining to each task. CLIPORT can even learn one multi-task model for all 10 tasks that achieves better or comparable performance to single-task models. Similarly, we demonstrate our approach on a Franka Panda manipulator with one multi-task model for 9 real-world tasks (k-o; only 5 shown) trained with just 179 image-action pairs.

To this end, we propose the first framework that combines the best of both worlds: end-to-end learning for fine-grained manipulation with the multi-goal and multi-task generalization capabilities of vision-language grounding systems. We introduce a two-stream architecture for manipulation with **semantic** and **spatial** pathways broadly inspired by (or vaguely analogous to) the two-stream hypothesis in cognitive psychology [16, 17, 18]. Specifically, we present CLIPORT, a language-conditioned imitation-learning agent that integrates the semantic understanding (*what*) of CLIP [1] with the spatial precision (*where*) of Transporter [2]. Transporter has been applied to a wide range of rearragement tasks from industrial packing [2] to manipulating deformable objects [6]. The key insight of the approach is formulating tabletop manipulation as a series of pick-and-place affordance predictions, where the objective is to *detect actions* rather than *detect objects* and then learn a policy. This action-centric approach to perception [19] is data efficient and effective at circumventing the need for explicit "objectness" in learnt representations. However, Transporter is a tabula rasa system that learns all visual representations from scratch and so every new goal or task requires collecting a new set of demonstrations. To address this problem, we bake in a strong semantic prior while learning policies. We condition our **semantic** stream with visual and language-goal features from a pre-trained CLIP model [1]. Since CLIP is pre-trained to align image and language features from millions of image-caption pairs from the internet, it provides a powerful prior for grounding semantic concepts that are common across tasks like categories, parts, shapes, colors, texts, and other visual attributes, all without a top-down pipeline that requires bounding boxes or instance segmentations [13, 14, 15, 20]. This allows us to formulate tabletop rearrangement as a series of language-conditioned affordance predictions, a predominantly vision-based inference problem, and thus benefit from the strengths of data-driven paradigms like scale and generalization.

To study these benefits, we conduct large-scale experiments in the Ravens [2] framework with a simulated suction-gripper robot. We propose 10 language-conditioned tasks with 1000s of unique instances per task that require both semantic and spatial reasoning (see Figure 1 a-j). CLIPORT is not only effective at solving these tasks, but surprisingly, it can even learn a multi-task model for all 10 tasks that achieves better or comparable performance to single-task models. Further, our evaluations indicate that our multi-task model can effectively transfer attributes like "pink block" across tasks, having never seen pink blocks or the word 'pink' in the context of the evaluation task. We also demonstrate our approach on a Franka Panda manipulator with one multi-task model for 9 real-world tasks trained with just 179 image-action pairs (see Figure 1 k-o).

In summary, our contributions are as follows:

- An extended **benchmark of language-grounding tasks** for manipulation in Ravens [2].
- **Two-stream architecture** for using internet pre-trained vision-language models for conditioning precise manipulation policies with language goals.
- **Empirical results** on a broad range of manipulation tasks, including multi-task models, validated with real-robot experiments.

The benchmark, code, and pre-trained models are available at: `cliport.github.io`.

## 2  Related Work

**Vision-based Manipulation.** Traditionally, perception for manipulation has centered around object detectors, segmentors, and pose estimators [21, 22, 23, 24, 25, 26]. These methods cannot handle deformable objects, granular media, or generalize to unseen objects without object-specific training data. Alternatively, dense descriptors [27, 28, 29] and keypoint representations [30, 31, 32] forgo segmentation and pose representations, but do not reason about sequential actions and struggle to represent scenes with variable numbers of objects. On the other hand, end-to-end perception-to-action models can learn precise sequential policies [2, 4, 6, 33, 34, 35], but these methods have limited understanding of semantic concepts and rely on goal-images to condition policies. In contrast, Yen-Chen et. al [36] showed that pre-training on semantic tasks like classification and segmentation helps in improving efficiency and generalization of grasping predictions.

**Semantic Models.** With the advent of large-scale models [37, 38, 39], a number of methods for learning joint vision and language representations have been proposed [13, 14, 15, 20, 40]. However, these methods are restricted to bounding boxes or instance segmentations, which make them inapplicable for detecting things like piles of coffee beans or squares on a chessboard. Alternatively, works in contrastive learning forgo top-down object-detection and learn continuous representations by pre-training on unlabeled data [11, 12]. Recently, CLIP [1] applied a similar approach to align vision and language representations by training on millions of image-caption pairs from the internet.

**Language Grounding for Robotics.** Several works have proposed systems for instructing robots with natural language [7, 8, 9, 10, 41, 42, 43, 44, 45, 46, 47]. However, these methods use disentangled pipelines for perception and action with the language primarily being used to guide the perception. As such, these pipelines lack the spatial precision necessary for tasks like folding cloths. Recently, Lynch et. al [48] proposed an end-to-end system for grounding language in continuous control, but it requires several hours of human teleoperation data for a single simulated desk setting.

**Two-Stream Architectures** are prevalent in action-recognition networks [49, 50, 51] and audio-recognition systems [52, 53]. In robotics, Zeng et. al [54] and Jang et. al [55] have proposed two-stream pipelines for affordance predictions of novel objects. The former requires goal-images and the latter is restricted to one-step grasps with single-category goals. In contrast, our framework provides a rich and intuitive interface with composable language commands for sequential tasks.

## 3  CLIPORT

CLIPORT is an imitation-learning agent based on four key principles: (1) Manipulation through a two-step primitive where each action involves a start and final end-effector pose. (2) Visual representations of actions that are equivariant to translations and rotations [56, 57]. (3) Two separate pathways for semantic and spatial information. (4) Language-conditioned policies for specifying goals and also transferring concepts across tasks. Combining (1) and (2) from Transporter with (3) and (4) allows us to achieve generalizable policies that go beyond just imitating demonstrations.

Section 3.1 describes the problem formulation, gives an overview of Transporter [2], and presents our language-conditioned model. Section 3.2 provides details on the training approach.

### 3.1  Language-Conditioned Manipulation

We consider the problem of learning a goal-conditioned policy $\pi$ that outputs actions $\mathbf{a}_t$ given input $\gamma_t = (\mathbf{o}_t, \mathbf{l}_t)$ consisting of a visual observation $\mathbf{o}_t$ and an English language instruction $\mathbf{l}_t$:

$$\pi(\boldsymbol{\gamma}_t) = \pi(\mathbf{o}_t, \mathbf{l}_t) \rightarrow \mathbf{a}_t = (\mathcal{T}_{\text{pick}}, \mathcal{T}_{\text{place}}) \in \mathcal{A} \tag{1}$$

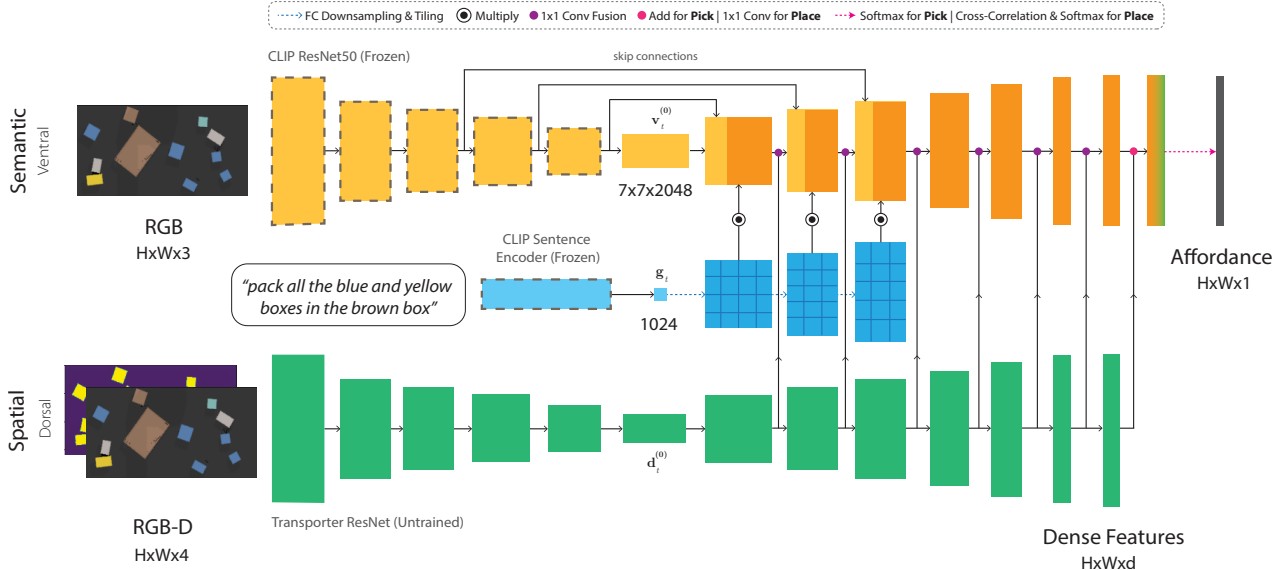

*Figure 2.* **CLIPORT Two-Stream Architecture.** An overview of the **semantic** and **spatial** streams. The **semantic** stream uses a frozen CLIP ResNet50 [1] to encode RGB input, and its decoder layers are conditioned with tiled language features from the CLIP sentence encoder. The **spatial** stream encodes RGB-D input, and its decoder layers are laterally fused with the **semantic** stream. The final output is a map of dense pixelwise features that is used for pick or place affordance predictions. This same two-stream architecture is used in all 3 Fully-Convolutional-Networks $f_{\text{pick}}$, $\Phi_{\text{query}}$, and $\Phi_{\text{key}}$ with $f_{\text{pick}}$ is used to predict pick actions, and $\Phi_{\text{query}}$ and $\Phi_{\text{key}}$ are used to predict place actions. See Appendix C for the exact architecture.

The actions $\mathbf{a} = (\mathcal{T}_{\text{pick}}, \mathcal{T}_{\text{place}})$ specify the end-effector pose for picking and placing, respectively. We consider tabletop tasks where $\mathcal{T}_{\text{pick}}, \mathcal{T}_{\text{place}} \in \mathbf{SE}(2)$. The visual observation $\mathbf{o}_t$ is a top-down orthographic RGB-D reconstruction of the scene where each pixel corresponds to a point in 3D space. The language instruction $\mathbf{l}_t$ either specifies step-by-step instructions e.g. *"pack the scissors"* $\rightarrow$ *"pack the purple tape"* $\rightarrow$ etc., or a single goal description for the whole task e.g *"pack all the blue and yellow boxes in the brown box"*. See Figure 4 for specific examples.

We assume access to a dataset $\mathcal{D} = \{\zeta_1, \zeta_2, \ldots, \zeta_n\}$ of $n$ expert demonstrations with associated discrete-time input-action pairs $\zeta_i = \{(\mathbf{o}_1, \mathbf{l}_1, \mathbf{a}_1), (\mathbf{o}_2, \mathbf{l}_2, \mathbf{a}_2), \ldots\}$ where $\mathbf{a}_t = (\mathcal{T}_{\text{pick}}, \mathcal{T}_{\text{place}})$ corresponds to expert pick-and-place coordinates at timestep $t$. These expert demonstrations are used to supervise the policy $\pi$.

**Transporter for Pick-and-Place.** The policy $\pi$ is trained with Transporter [2] to perform spatial manipulation. The model first (i) attends to a local region to decide where to pick, then (ii) computes a placement location by finding the best match through cross-correlation of deep visual features.

Following Transporter [2, 6], the policy $\pi$ is composed of two action-value modules (Q-functions): The pick module $\mathcal{Q}_{\text{pick}}$ decides where to pick, and conditioned on this pick action the place module $\mathcal{Q}_{\text{place}}$ decides where to place. These modules are implemented as Fully-Convolutional-Networks (FCNs) that are translationally equivariant by design. As we will describe in more detail below, we extend these networks to two-stream architectures that can handle language input. The pick FCN $f_{\text{pick}}$ takes input $\gamma_t = (\mathbf{o}_t, \mathbf{l}_t)$ and outputs a dense pixelwise prediction $\mathcal{Q}_{\text{pick}} \in \mathbb{R}^{H \times W}$ of action-values, where are used to predict the pick action $\mathcal{T}_{\text{pick}}$:

$$\mathcal{T}_{\text{pick}} = \underset{(u,v)}{\arg\max}\ \mathcal{Q}_{\text{pick}}((u,v)|\gamma_t) \tag{2}$$

Since $\mathbf{o}_t$ is an orthographic heightmap, each pixel location $(u, v)$ can be mapped to a 3D picking location using the known camera calibration. $f_{\text{pick}}$ is trained in a supervised manner to predict the pick action $\mathcal{T}_{\text{pick}}$ that imitates the expert demonstration with the specified language instruction at timestep $t$.

The second FCN $\Phi_{\text{query}}$ takes in $\gamma_t[\mathcal{T}_{\text{pick}}]$, which is a $c \times c$ crop of $\mathbf{o}_t$ centered at $\mathcal{T}_{\text{pick}}$ along with the language instruction $\mathbf{l}_t$, and outputs a query feature embedding of shape $\mathbb{R}^{c \times c \times d}$. The third FCN $\Phi_{\text{key}}$ consumes the full input $\gamma_t$ and outputs a key feature embedding of shape $\mathbb{R}^{H \times W \times d}$. The place action-values $\mathcal{Q}_{\text{place}}$ are then computed by cross-correlating the query and key features:

$$\mathcal{Q}_{\text{place}}(\Delta\tau|\gamma_t, \mathcal{T}_{\text{pick}}) = \big(\Phi_{\text{query}}(\gamma_t[\mathcal{T}_{\text{pick}}]) * \Phi_{\text{key}}(\gamma_t)\big)[\Delta\tau] \tag{3}$$

where $\Delta\tau \in \boldsymbol{SE}(2)$ represents a potential placement pose. Since $\mathbf{o}_t$ is an orthographic heightmap, rotations in the placement pose $\Delta\tau$ can be captured by stacking $k$ discrete angle rotations of the crop before passing it through the query network $\Phi_{\text{query}}$. Then $\mathcal{T}_{\text{place}} = \text{argmax}_{\Delta\tau} \mathcal{Q}_{\text{place}}(\Delta\tau | \gamma_t, \mathcal{T}_{\text{pick}})$, where the place module is trained to imitate the placements in the expert demonstrations. For all models, we use $c = 64$, $k = 36$ and $d = 3$. As in Transporter [2, 6], our framework can be extended to handle any motion primitive like pushing, sliding, etc. that can be parameterized by two end-effector poses at each timestep. For more details, we refer the reader to the original paper [2].

**Two-Stream Architecture.** In CLIPORT, we extend the network architecture of all three FCNs $f_{\text{pick}}, \Phi_{\text{query}}$ and $\Phi_{\text{key}}$ from Transporter [2] to allow for language input and reasoning about high-level semantic concepts. We extend the FCNs to two-pathways: **semantic** (ventral) and **spatial** (dorsal). The **semantic** stream is conditioned with language features at the bottleneck and fused with intermediate features from the **spatial** stream. See Figure 2 for an overview of the architecture.

The **spatial** stream is identical to the ResNet architecture in Transporter – a tabula rasa network that takes in RGB-D input $\mathbf{o}_t$ and outputs dense features through an hourglass encoder-decoder model. The **semantic** stream uses a frozen pre-trained CLIP ResNet50 [1] to encode the RGB input[2] $\tilde{\mathbf{o}}_t$ up until the penultimate layer $\tilde{\mathbf{o}}_t \rightarrow \mathbf{v}_t^{(0)} : \mathbb{R}^{7 \times 7 \times 2048}$, and then introduces decoding layers that upsample the feature tensors to mimic the **spatial** stream $\mathbf{v}_t^{(l-1)} \rightarrow \mathbf{v}_t^{(l)} : \mathbb{R}^{h \times w \times C}$ at each layer $l$.

The language instruction $\mathbf{l}_t$ is encoded with CLIP's Transformer-based sentence encoder to produce a goal encoding $\mathbf{l}_t \rightarrow \mathbf{g}_t : \mathbb{R}^{1024}$. This goal encoding $\mathbf{g}_t$ is downsampled with fully-connected layers to match the channel dimension $C$ and tiled to match the spatial dimensions of the decoder features such that $\mathbf{g}_t \rightarrow \mathbf{g}_t^{(l)} : \mathbb{R}^{h \times w \times C}$. The decoder features are then conditioned with the tiled goal features through an element-wise product $\mathbf{v}_t^{(l)} \odot \mathbf{g}_t^{(l)}$ (Hadamard product). Since CLIP was trained with contrastive loss on the dot-product alignment between pooled image features and language encodings, the element-wise product allows us to use this alignment while the tiling preserves the spatial dimensions of the visual features. This language conditioning is repeated for three subsequent layers after the bottleneck inspired by LingUNet [58]. We also add skip connections to these layers from the CLIP ResNet50 encoder to utilize different levels of semantic information from shapes to parts to object-level concepts [59]. Finally, following existing two-stream architectures in video-action recognition [51], we add lateral connections from the **spatial** stream to the **semantic** stream. These connections involve concatenating two feature tensors and applying $1 \times 1$ conv to reduce the channel dimension $[\mathbf{v}_t^{(l)} \odot \mathbf{g}_t^{(l)}; \mathbf{d}_t^{(l)}] : \mathbb{R}^{h \times w \times C_{\mathbf{v}} + C_{\mathbf{d}}} \rightarrow \mathbb{R}^{h \times w \times C_{\mathbf{v}}}$, where $\mathbf{v}_t^{(l)}$ and $\mathbf{d}_t^{(l)}$ are the **semantic** and **spatial** tensors at layer $l$, respectively. For the final fusion of dense features, `addition` for $f_{\text{pick}}$ and $1 \times 1$ `conv` fusion for $\Phi_{\text{query}}$ and $\Phi_{\text{key}}$ worked the best empirically. See Appendix C for details on the exact architecture.

### 3.2 Implementation Details

**Training from demonstrations.** Similar to Transporter [2] we train CLIPORT through imitation learning from a set of expert demonstrations $\mathcal{D} = \{\zeta_1, \zeta_2, \ldots, \zeta_n\}$ consisting of discrete-time input-action pairs $\zeta_i = \{(\mathbf{o}_1, \mathbf{l}_1, \mathbf{a}_1), (\mathbf{o}_2, \mathbf{l}_2, \mathbf{a}_2), \ldots\}$. During training, we randomly sample an input-action pair from the dataset and supervise the model end-to-end with one-hot pixel encodings of demonstration actions $Y_{\text{pick}} : \mathbb{R}^{H \times W \times k}$ and $Y_{\text{place}} : \mathbb{R}^{H \times W \times k}$ with $k$ discrete rotations. In simulated experiments with the suction-gripper, we use $k = 1$ for pick actions and $k = 36$ for place actions. The model is trained with cross-entropy loss: $\mathcal{L} = -\mathbb{E}_{Y_{\text{pick}}}[\log \mathcal{V}_{\text{pick}}] - \mathbb{E}_{Y_{\text{place}}}[\log \mathcal{V}_{\text{place}}]$ where $\mathcal{V}_{\text{pick}} = \text{softmax}(\mathcal{Q}_{\text{pick}}((u, v) | \gamma_t))$ and $\mathcal{V}_{\text{place}} = \text{softmax}(\mathcal{Q}_{\text{place}}((u', v', \omega') | \gamma_t, \mathcal{T}_{\text{pick}}))$. Compared to the original Transporter models that were trained for 40K iterations, we train our models for 200K iterations (with data augmentation; see Appendix E) to account for additional semantic variation in tasks – randomized colors, shapes, objects. All models are trained on a single commodity GPU for 2 days with a batch size of 1.

**Training multi-task models.** Multi-task training is nearly identical to single-task training except for the sampling of training data. First, we randomly sample a task, and then select a random input-action pair from that task in the dataset. Using this strategy, all tasks are equally likely to be sampled but longer horizon tasks are less likely to reach full coverage of input-action pairs available in the dataset. To compensate for this, we train all multi-task models $3\times$ longer for 600K iterations or 6 GPU days.

---

[2]We cannot use depth information with CLIP since it was trained with RGB-only image-caption pairs from the internet.

# 4 Results

We perform experiments both in simulation and hardware aimed at answering the following questions: 1) How effective is the language-conditioned two-stream architecture for fine-grained manipulation compared to one-stream alternatives and other simpler baselines? 2) Is it possible to train a multi-task model for all tasks, and how well does it perform and generalize? 3) How well do these models generalize to seen and unseen semantic attributes like colors, shapes, and object categories?

## 4.1 Simulation Setup

**Environment.** All simulated experiments are based on a Universal Robot UR5e with a suction gripper. The setup provides a systematic and reproducible environment for evaluation, especially for benchmarking the ability to ground semantic concepts like colors and object categories. The input observation is a top-down RGB-D reconstruction from 3 cameras positioned around a rectangular table: one in the front, one on the left shoulder, and one on the right shoulder, all pointing towards the center. Each camera has a resolution of $640 \times 480$ and is noiseless.

**Language-Conditioned Manipulation Tasks.** We extend the Ravens benchmark [2] set in PyBullet [60] with 10 language-conditioned manipulation tasks. See Figure 1 for examples and Table 3 for challenges associated with each task. Each task instance is constructed by sampling a set of objects and attributes: poses, colors, sizes, and object categories. 8 of the 10 tasks have two variants, denoted by seen and unseen, depending on whether the task has unseen attributes (e.g. color) at test time. For colors: $\mathbb{T}_{\text{seen colors}} = \{\texttt{yellow, brown, gray, cyan}\}$ and $\mathbb{T}_{\text{unseen colors}} = \{\texttt{orange, purple, pink, white}\}$ with 3 overlapping colors $\mathbb{T}_{\text{all}} = \{\texttt{red, green, blue}\}$ used in both the seen and unseen spilts. For packing objects, we use 56 tabletop objects from the Google Scanned Objects dataset [61] and split them into 37 seen and 19 unseen objects. The language instructions are constructed from templates for simulated experiments, and human-annotated for real-world experiments. For more details about individual tasks, see Appendix A.

**Evaluation Metric.** We adopt the 0 (fail) to 100 (success) scores proposed in the Ravens benchmark [2]. The score assigns partial credit based on the task, e.g. $3/5 \Rightarrow 60.0$ for packing 3 out of 5 objects specified in the instructions, or $30/56 \Rightarrow 53.6$ for pushing 30 out of 56 particles into the correct zone. See Appendix A for the specific evaluation metric used in each task. During an evaluation episode, an agent keeps interacting with the scene until an oracle indicates task-completion. We report scores on 100 evaluation runs for agents trained with $n = 1, 10, 100, 1000$ demonstrations.

## 4.2 Simulation Results

Table 1 presents results from our large-scale experiments in Ravens [2] and Figure 3 summarizes these results with average scores across seen and unseen splits.

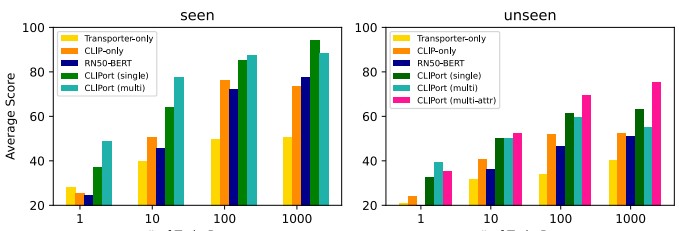

*Figure 3.* Average scores across seen and unseen splits for all tasks in Table 1.

**Baseline Methods.** To study the effectiveness of our two-stream architecture, we broadly compare against two baselines: Transporter-only and CLIP-only. Transporter-only is the original Transporter [2], or equivalently, the **spatial** stream of CLIPORT with RGB-D input. Although Transporter-only does not receive any language goals, it shows what can be achieved through chance by exploiting the most likely actions seen during training. On the other hand, CLIP-only is just the **semantic** stream of CLIPORT with RGB and language input. CLIP-only shows what can be achieved by fine-tuning a pre-trained semantic model for manipulation without spatial information, particularly depth.

**Two-Stream Performance.** Figure 3 (seen) captures the essence of our main claims. The performance of Transporter-only saturates at $50\%$ since it doesn't use the language instruction to ground the desired goal. CLIP-only does have a goal, but lacks the spatial precision to go the last mile and thus saturates at $76\%$. Only CLIPORT (single) achieves more than $90\%$, which indicates that both the semantic and spatial streams are crucial for fine-grained manipulation. Further, CLIPORT (single) achieves $86\%$ on most tasks with just 100 demonstrations, showcasing its efficiency.

In addition to these baselines, we present various ablations and alternative one-stream and two-stream models in Appendix F. To briefly summarize these results, CLIP is essential for few-shot

| Method | packing-box-pairs seen-colors | | | | packing-box-pairs unseen-colors | | | | packing-seen-google objects-seq | | | | packing-unseen-google objects-seq | | | | packing-seen-google objects-group | | | | packing-unseen-google objects-group | | | |
|---|---|---|---|---|---|---|---|---|---|---|---|---|---|---|---|---|---|---|---|---|---|---|---|---|
| | 1 | 10 | 100 | 1000 | 1 | 10 | 100 | 1000 | 1 | 10 | 100 | 1000 | 1 | 10 | 100 | 1000 | 1 | 10 | 100 | 1000 | 1 | 10 | 100 | 1000 |
| Transporter-only [2] | 44.2 | 55.2 | 54.2 | 52.4 | 34.6 | 48.7 | 47.2 | 54.1 | 26.2 | 39.7 | 45.4 | 46.3 | 19.9 | 29.8 | 28.7 | 37.3 | 60.0 | 54.3 | 61.5 | 59.9 | 46.2 | 54.7 | 49.8 | 52.0 |
| CLIP-only | 38.6 | 69.7 | 88.5 | 87.1 | 33.0 | 65.5 | 68.8 | 61.2 | 29.1 | 67.9 | **89.3** | 95.8 | 37.1 | 49.4 | 60.4 | 57.8 | 52.5 | 62.0 | **89.6** | **92.7** | 43.4 | 65.9 | 73.1 | 70.0 |
| RN50-BERT | 36.2 | 64.0 | **94.7** | 90.3 | 31.4 | 52.7 | 65.6 | **72.1** | 32.9 | 48.4 | 87.9 | 94.0 | 29.3 | 48.5 | 48.3 | 56.1 | 46.4 | 52.9 | 76.5 | 86.4 | 43.2 | 52.0 | 66.3 | 73.7 |
| CLIPORT (single) | 51.6 | 82.9 | 92.7 | **98.2** | 45.6 | 65.3 | 68.6 | 71.5 | 14.8 | 59.5 | 86.8 | **96.2** | 27.2 | 50.0 | 65.5 | **71.9** | 52.7 | 67.0 | 84.1 | 94.0 | 61.5 | 66.2 | 78.4 | **81.5** |
| CLIPORT (multi) | **66.8** | **88.6** | 94.1 | 96.6 | **59.0** | **69.7** | **76.2** | 71.4 | **41.6** | **78.4** | 85.0 | 84.4 | **40.7** | **51.1** | **65.8** | 70.3 | **71.3** | **84.6** | 89.6 | 88.3 | **68.4** | **69.6** | **78.4** | 80.3 |
| CLIPORT (multi-attr) | – | – | – | – | *46.2* | *72.0* | *86.2* | *80.3* | – | – | – | – | *35.4* | *45.1* | *78.9* | *87.4* | – | – | – | – | *48.6* | *69.3* | *84.8* | *89.1* |

| Method | stack-block-pyramid seq-seen-colors | | | | stack-block-pyramid seq-unseen-colors | | | | separating-piles seen-colors | | | | separating-piles unseen-colors | | | | towers-of-hanoi seq-seen-colors | | | | towers-of-hanoi seq-unseen-colors | | | |
|---|---|---|---|---|---|---|---|---|---|---|---|---|---|---|---|---|---|---|---|---|---|---|---|---|
| | 1 | 10 | 100 | 1000 | 1 | 10 | 100 | 1000 | 1 | 10 | 100 | 1000 | 1 | 10 | 100 | 1000 | 1 | 10 | 100 | 1000 | 1 | 10 | 100 | 1000 |
| Transporter-only [2] | 4.5 | 2.3 | 5.2 | 4.5 | 3.0 | 4.0 | 2.3 | 5.8 | 42.7 | 52.3 | 42.0 | 48.4 | 41.2 | 49.2 | 44.7 | 52.3 | 25.4 | 67.9 | 98.0 | 99.9 | 24.3 | 44.6 | 71.7 | 80.7 |
| CLIP-only | 6.3 | 28.7 | 55.7 | 54.8 | 2.0 | 12.2 | 18.3 | 19.5 | 43.5 | 55.0 | 84.9 | 90.2 | **59.9** | 49.6 | 73.0 | 71.0 | 9.4 | 52.6 | 88.6 | 45.3 | 24.7 | 47.0 | 67.0 | 58.0 |
| RN50-BERT | 5.3 | 35.0 | 89.0 | 97.5 | 6.2 | 12.2 | 21.5 | 30.7 | 31.8 | 47.8 | 46.5 | 46.5 | 33.4 | 44.4 | 41.3 | 44.9 | 28.0 | 66.1 | 91.3 | 92.1 | 17.4 | 75.1 | 85.3 | 89.3 |
| CLIPORT (single) | 28.3 | 64.7 | 93.3 | **98.8** | 13.7 | 24.3 | 31.2 | **41.3** | **54.5** | 59.5 | **93.1** | **98.0** | 47.2 | 51.0 | **76.6** | **75.2** | 59.4 | 92.9 | 97.4 | **100** | 56.1 | **89.7** | **95.9** | **99.4** |
| CLIPORT (multi) | **33.5** | **75.3** | **96.8** | 96.5 | **23.3** | **26.8** | **31.7** | 22.2 | 48.9 | **72.4** | 90.3 | 89.0 | 56.6 | **62.6** | 64.9 | 62.8 | **61.6** | **96.3** | **98.7** | 98.1 | **60.1** | 65.6 | 76.7 | 68.7 |
| CLIPORT (multi-attr) | – | – | – | – | *15.5* | *51.5* | *59.3* | *79.8* | – | – | – | – | *49.9* | *51.8* | *48.2* | *59.8* | – | – | – | – | *56.7* | *78.0* | *88.3* | *96.9* |

| Method | align-rope | | | | packing-unseen-shapes | | | | assembling-kits-seq seen-colors | | | | assembling-kits-seq unseen-colors | | | | put-blocks-in-bowls seen-colors | | | | put-blocks-in-bowls unseen-colors | | | |
|---|---|---|---|---|---|---|---|---|---|---|---|---|---|---|---|---|---|---|---|---|---|---|---|---|
| | 1 | 10 | 100 | 1000 | 1 | 10 | 100 | 1000 | 1 | 10 | 100 | 1000 | 1 | 10 | 100 | 1000 | 1 | 10 | 100 | 1000 | 1 | 10 | 100 | 1000 |
| Transporter-only [2] | 6.9 | 30.6 | 33.1 | 51.5 | 16.0 | 20.0 | 22.0 | 22.0 | 5.8 | 11.6 | 28.6 | 29.6 | 7.8 | 17.6 | 25.6 | 28.4 | 16.8 | 33.3 | 62.7 | 64.7 | 11.7 | 17.2 | 14.8 | 18.7 |
| CLIP-only | 13.4 | 48.7 | 70.4 | 70.7 | 13.0 | 28.0 | **44.0** | **50.0** | 0.8 | 9.2 | 19.8 | 23.0 | 2.0 | 4.6 | 10.8 | 19.8 | 23.5 | 60.2 | 93.5 | 97.7 | 11.2 | 34.2 | 33.2 | 44.5 |
| RN50-BERT | 3.1 | 25.0 | 63.8 | 57.1 | 19.0 | 25.0 | 32.0 | 44.0 | 2.2 | 5.6 | 11.6 | 21.8 | 1.6 | 6.4 | 10.4 | 18.4 | 13.8 | 44.5 | 81.2 | 91.8 | 16.2 | 23.0 | 30.3 | 23.8 |
| CLIPORT (single) | **20.1** | **77.4** | **85.6** | **95.4** | 21.0 | 26.0 | 40.0 | 37.0 | **12.2** | 17.8 | **47.0** | **66.6** | **16.2** | 18.0 | **35.4** | **34.8** | 23.5 | 68.3 | 92.5 | **100** | 18.0 | 35.3 | 37.3 | 25.0 |
| CLIPORT (multi) | 19.6 | 49.3 | 82.4 | 74.9 | **25.0** | **35.0** | 37.0 | 31.0 | 11.4 | **34.8** | 46.2 | 52.4 | 7.8 | **21.6** | 29.0 | 25.4 | **54.0** | **90.2** | **99.5** | **100** | **32.0** | **48.8** | **55.3** | **45.8** |
| CLIPORT (multi-attr) | – | – | – | – | – | – | – | – | – | – | – | – | *7.6* | *10.4* | *43.8* | *34.6* | – | – | – | – | *23.0* | *41.8* | *66.5* | *75.7* |

Table 1. **Language-Conditioned Test Results.** Task success scores (mean %) from 100 evaluation instances vs. # of training demonstrations (1, 10, 100, or 1000). The challenges pertaining to each task are described in Appendix A. CLIPORT (single) models are trained on seen splits, and evaluated on both seen and unseen splits. CLIPORT (multi) models are trained on seen splits of all 10 tasks with $1\mathbb{T}$, $10\mathbb{T}$, $100\mathbb{T}$, and $1000\mathbb{T}$ demonstrations where $\mathbb{T} = 10$. CLIPORT (multi-attr) indicate CLIPORT (multi) models trained on seen-and-unseen splits from all tasks *except* for that one particular heldout task, for which it is trained only the seen split. See Figure 3 for an overview with average scores.

learning (i.e. $n \geq 10$) in lieu of **semantic** stream alternatives like ImageNet-trained ResNet50 [62] with BERT [38]. Image-goal models outperform CLIPORT (single) in packing Google objects, but this is only because they do not have to solve the language-grounding problem.

**Multi-Task Performance.** In realistic scenarios, we want the robot to be capable of any task, not just one task. We investigate this through CLIPORT (multi) in Table 1 with one multi-task model trained on all 10 tasks. CLIPORT (multi) models are trained only on seen-splits of tasks, so an unseen attribute like 'pink' is consistent throughout single and multi-task settings. Surprisingly, CLIPORT (multi) outperforms single-task CLIPORT (single) models in $41/72 = 57\%$ of the evaluations in Table 1. This trend is also evident in Figure 3 (seen), especially in instances with 100 demonstrations or less. Although CLIPORT (multi) is trained on more diverse data from other tasks, both CLIPORT (multi) and CLIPORT (single) have access to the same amount of data *per* task. This supports our premise that language is a strong conditioning mechanism for reusing concepts from other tasks without learning them from scratch. It also validates a trait of data-driven approaches where training on lots of diverse data leads to more robust and generalizable representations [1, 63]. However, CLIPORT (multi) performs worse on longer-horizon tasks like align-rope. We hypothesize that this is because longer-horizon tasks get less coverage of input-action pairs in the dataset. Future works could use better sampling methods that balance tasks according to their average time horizon.

**Generalizing to Unseen Attributes.** Tasks that require generalizing to novel colors, shapes, and objects are more difficult and all our agents achieve relatively lower performance on these tasks, as shown in Figure 3 (unseen). However, CLIPORT (single) models do substantially better than chance, i.e., Transporter-only. The lower performances are due to the difficulty of grounding unseen attributes such as 'pink' and 'orange' in the language instruction "put the pink block on the orange bowl", when the agent has never encountered words 'orange', 'pink' or their corresponding visual characteristics in the context of the physical environment. Although pre-trained CLIP has been exposed to the attribute 'pink', it could correspond to different concepts in the physical setting depending on factors like lighting condition, and thus requires at least few examples to condition the trainable **semantic** decoder layers. Additionally, we notice that CLIPORT (single) is also less prone to overfitting compared to Transporter-only. As evidenced in towers-of-hanoi-seq-unseen-colors task in Table 1, Transporter-only suffers from a performance drop because of rings with unseen colors despite the fact that Tower of Hanoi can be solved without attending to the colors and simply focusing on the ring size. We hypothesize that since CLIP was trained on diverse internet data, it enables our agent to focus on task-relevant concepts while ignoring irrelevant aspects of the task.

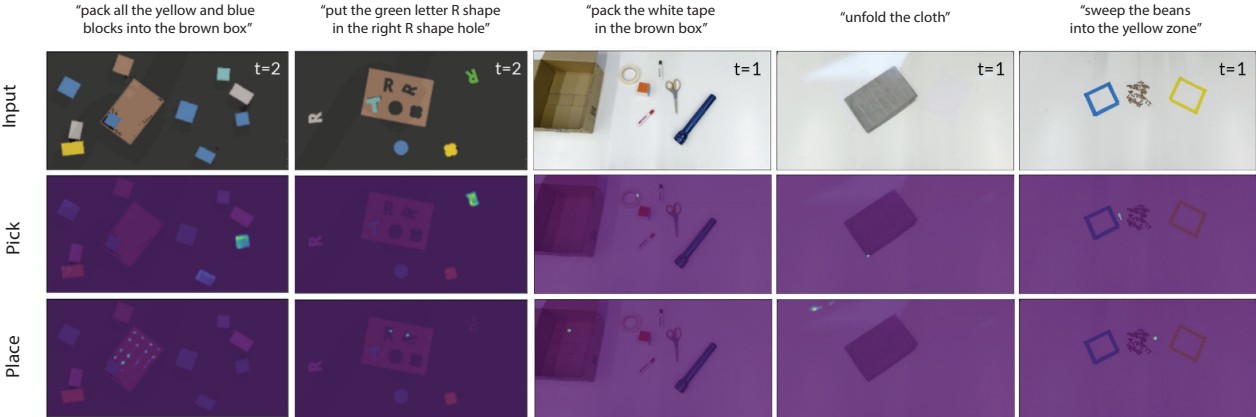



"pack all the yellow and blue blocks into the brown box" · "put the green letter R shape in the right R shape hole" · "pack the white tape in the brown box" · "unfold the cloth" · "sweep the beans into the yellow zone"



*Figure 4.* Affordance predictions from CLIPORT (multi) models in sim (left two) and real settings (right three). More examples in Appendix H.

**Transferring Attributes across Tasks.** One solution for dealing with unseen attributes is to explicitly learn these attributes from other tasks. We study this with CLIPORT (multi-attr) in Table 1 and Figure 3 (unseen). For these models, CLIPORT (multi) is trained on both seen-and-unseen splits from all tasks *except* for the task being evaluated on, for which it was only trained on the seen split. As such, this evaluation measures whether having seen pink blocks in put-blocks-in-bowl-unseen-colors helps solve "pack all the pink and cyan boxes" in packing-box-pairs-unseen-colors. Results indicate that such explicit transfers result in significant improvements. For instance, on the put-blocks-in-bowls-unseen-colors task for $n = 1000$, CLIPORT (multi)'s performance increases from $45.8$ to $75.7$.

### 4.3 Real-Robot Experiments

We validated our results in hardware with a Franka Panda manipulator. See Appendix D for setup details. Table 2 reports success rates for a multi-task model trained and evaluated on 9 real-world tasks. Due to COVID restrictions, we could not conduct large-scale user-studies, so we report on small train (5-10 demos) and test sets (5-10 runs) per task. Overall, CLIPORT (multi) is effective at few-shot learning with just 179 samples, and the performances roughly correspond to those in simulated experiments, with simple block ma-

| Task | # Train (Samples) | # Test | Succ. % |
|---|---|---|---|
| Stack Blocks | 5 (13) | 10 | 70.0 |
| Put Blocks in Bowl | 5 (10) | 10 | 65.0 |
| Pack Objects | 10 (31) | 10 | 60.0 |
| Move Rook | 4 (29) | 10 | 70.0 |
| Fold Cloth | 9 (9) | 10 | 57.0 |
| Read Text | 2 (26) | 10 | 55.0 |
| Loop Rope | 4 (12) | 10 | 60.0 |
| Sweep Beans | 5 (23) | 5 | 60.6 |
| Pick Cherries | 4 (26) | 5 | 75.0 |

*Table 2.* Success rates (%) of a multi-task model trained an evaluated 9 real-world tasks (see Figure 1). Samples indicate total image-action pairs, e.g 1 in Figure 9.

nipulation tasks achieving $\sim 70\%$. We estimate that for more robust real-world performance at least 50 to 100 training demonstrations are necessary, as evident in Figure 3. Interestingly, we observed that the model sometimes exploits biases in the training data instead of learning to ground instructions. For instance, in Put Blocks in Bowl, the training set consisted of only one datapoint on "yellow blocks" being placed inside a "blue bowl". This made it difficult to condition the model to place "yellow blocks" in non-blue bowls. But instances with just one or two examples where a colored block went to different colored bowls was sufficient to make the model pay attention to the language. In summary, unbiased datasets containing both a good coverage of expected skills and invariances, and a decent number of training demonstrations, are crucial for good real-world performance.

## 5 Conclusion

We introduced CLIPORT, an end-to-end framework for language-conditioned fine-grained manipulation. Our experiments, specifically with multi-task models, indicate that data-driven approaches to generalization have yet to be fully-exploited in robotics. Coupled with the right action abstraction and spatio-semantic priors, end-to-end methods can quickly learn new skills without requiring top-down pipelines that need task-specific engineering.

While CLIPORT can solve a range of tabletop tasks, extending it to dexterous 6-DOF manipulation that goes beyond the two-step primitive remains a challenge. As such, it cannot handle complex partially-observable scenes, or output continuous control for multi-fingered hands, or predict task-completion (see Appendix I for an extended discussion). But overall, we are excited by the confluence of data and structural priors for building scalable and generalizable robotic systems.

**Acknowledgments**

All simulated experiments were facilitated through the Hyak computing cluster funded by the STF at the University of Washington. We thank Mohak Bhardwaj for help with the Franka setup at UW. We are also grateful to our colleagues Chris Xie, Jesse Thomason, and Valts Blukis for providing feedback on the initial draft. This work was funded in part by ONR under award #1140209-405780.

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
