# OpenReview forum: "CLIPort: What and Where Pathways for Robotic Manipulation"
_robot-learning.org/CoRL/2021/Conference — CoRL2021 Poster_

### Official Review · Reviewer_HUjp · 2021-07-02

**Originality:** Fair
**Technical Quality:** Very Good
**Clarity Of Presentation:** Good
**Impact:** 4

**Recommendation:**

Weak Accept: I recommend accepting the paper, but will not argue for my recommendation if the majority of other reviewers have a different opinion.

**Summary:**

The paper proposes a language-conditioned modification of Transporter (Zeng’20). Pretrained CLIP (Radford’21) modules are used for the visual and the language stream with fine-tuning on language-annotated demonstrations. The proposed model succeeds on a range of tabletop spatial manipulation tasks in simulation and real world. The model outperforms chance performance (i.e. no language input) on most tasks. Using depth as an additional input somewhat improves performance. Multi-task training improves data efficiency, especially when trained on larger datasets. The model somewhat outperforms standard pretraining of the vision and language models separately (e.g. ImageNet and BERT models), because it leverages joint vision-language pretraining (see App. Tab 4).


**Issues:**


See weaknesses

**Reviewer Expertise:**

Very good: Comprehensive knowledge of the area

**Strengths And Weaknesses:**


Strengths:
- The paper presents a promising system that can perform spatial manipulation tasks from language and visual input.
- Several interesting experiments and ablations on this system are presented.
- The proposed method leverages large-scale internet datasets via using the pretrained CLIP model, which is a promising direction for robotic control.

Weaknesses:
- As the paper notices on l319, the base Transporter architecture is quite limited in terms of tasks it can perform and it’s unclear whether the finding of this paper will generalize to more general architectures.
- The paper claims that “CLIP is essential for few-shot learning in lieu of alternatives like ImageNet-trained ResNet50 [56] with BERT [35].” That does not follow from the comparison in the App. Tab 4. In fact, ImageNet ResNet + BERT outperforms CLIP in the few-shot regime on two of the 4 tasks.
- In general, the experimental evaluation is highly confusing and lacks many baselines that one would want to see. For instance, the no pretraining baseline or the ImageNet ResNet + BERT baseline are the natural choice, but they are omitted in favor of chance performance, a minor architecture ablation, and an ablation that uses more data. More informative baselines are provided in the appendix. Perhaps just switching the baselines in the paper for the baselines in the appendix would solve this problem?



**Summary Of Recommendation:**


The paper attacks an interesting problem and presents a system that has many of the right parts to attack that problem. I am leaning towards accepting the paper, but the presentation of the baselines needs to be significantly improved for that.

---

> ### Author Response · Authors · 2021-08-23
> **Response to Reviewer HUJp – Part 1**
>
> _As the paper notices on l319, the base Transporter architecture is quite limited in terms of tasks it can perform and it’s unclear whether the finding of this paper will generalize to more general architectures._
>
> The authors agree that the Transporter architecture has some limitations in terms of the tasks it can perform since it relies on a top-down RGB-D heightmap and pick-and-place manipulation actions. However, the architecture exploits the spatial structure of tabletop manipulation problems to efficiently learn generalizable policies and has numerous advantages that we believe make this an interesting approach. First and foremost we would like to highlight the fact that the Transporter architecture (and by extension CLIPort) is able to accomplish precise manipulation tasks **in the real world** with a relatively small number of human demonstrations – the real-world experiments in our paper used a total of 179 image-action training pairs. This is order of magnitudes more efficient than other imitation learning approaches or alternatives like model-based or model-free reinforcement learning. By building off the Transporter architecture for CLIPort we are able to capture many of the benefits of Transporter while adding the benefits of language-conditioned manipulation.
>
> Here we give some specific examples of the aforementioned benefits of the Transporter Networks architecture in the context of a few specific manipulation tasks we consider in the paper. For a task like sweeping beans, popular approaches like learning visual dynamics models for model-based planning or reinforcement learning would be very difficult. Due to the stochastic nature of sweeping particle objects, forward models will often make blurry predictions of future states. In contrast, CLIPort can effectively sweep the beans into a specific zone, without compromising on the precision needed for detecting and manipulating every single bean, all without any supervision on bean segmentations. Similarly, while the supervised-learning setup in Transporter relies on an expert for training demonstrations, for some tasks this might be the safest way to learn a good policy. In the cherry-picking task, grabbing the fruit directly with the parallel gripper will squeeze the fruit and destroy it. In self-supervised and reinforcement learning settings, the robot will have to ruin a lot of cherries while taking random actions to eventually learn a useful policy, but in a supervised setting we can teach the model to always pick by the stem. And again, this cherry-pick task was just trained with 26 image-action pairs.

---

> > ### Author Response · Authors · 2021-08-23
> > **Response to Reviewer HUJp - Part 2**
> >
> > _The paper claims that “CLIP is essential for few-shot learning in lieu of alternatives like ImageNet-trained ResNet50 [56] with BERT [35].” That does not follow from the comparison in the App. Tab 4. In fact, ImageNet ResNet + BERT outperforms CLIP in the few-shot regime on two of the 4 tasks._
> >
> > We believe this is a misreading as reviewer R-HUjp is referring to “one-shot” results for n=1 packing-seen-google-object-seq and packing-unseen-google-object-seq. In the “few-shot” setting, i.e. n=10, the gains of CLIPort over RN50-BERT are 36 -> 69 for the seen task and 48 -> 60 for the unseen task. This might not be obvious to readers, accordingly we have updated the writing to make this few-shot reference more explicit.
> >
> > Note that performances for one-shot (n=1) packing-google-objects are more up to chance rather than anything else. The models have literally seen just a single demonstration with a couple of objects, and have to generalize to all other unseen instances of Google Scanned Objects in the dataset. So we can only draw meaningful conclusions from n>=10 for packing Google objects.
> >
> > Also, each of these 72 evaluations are an average of 100 randomized evaluations instances, which total to 32,900 individual tests in Table 1. So a difference in performance like 36 -> 69 is quantitatively substantial. Please see the videos at the bottom of [cliport.github.io](http://cliport.github.io) to get a sense of the complexity associated with each evaluation. This section contains a 5% subset of the 100 evaluation instances for all tasks.
> >
> > _In general, the experimental evaluation ... lacks many baselines that one would want to see. For instance, the no pretraining baseline or the ImageNet ResNet + BERT baseline are the natural choice, but they are omitted in favor of chance performance, a minor architecture ablation, and an ablation that uses more data...?_
> >
> > We have added RN50-BERT baselines for all tasks in Table 1 and Figure 3. CLIPort outperforms ImagetNet-trained ResNet50 with BERT in 68/72 = 94% of the evaluations in Table 1.
> >
> > We have added a no pre-training baseline (Untrained-Sem-Transporter) to Appendix Table 4. Pre-trained semantic-streams lead to substantial improvements in performance compared to untrained semantic-streams.
> >
> > We thank R-HUjp for suggesting these baselines. We believe these new results have strengthened our claims.
> >
> > Additionally we would like to note that the CLIPort (multi-attr) is more than just a CLIPort “ablation that uses more data.” It is designed to test whether CLIPort can transfer semantic attributes across tasks, which is demonstrated in the affirmative. We, the authors, are particularly excited about the multi-task results as a single trained model can be used to solve various tasks through just language commands.

---

> > > ### Comment · Reviewer_HUjp · 2021-08-23
> > > **Reviewer's response**
> > >
> > > Thank you for the updates. The updates address some of my concerns. Additionally, I second the concern of Hq8X that the proposed approach can only make limited use of language since it only addresses spatial manipulation tasks for which language commands are generally very simple. A more general approach that can fully leverage the language information by solving more general tasks would be preferable.
> > >
> > > I note that this perspective further invites a comparison to an object detection-pretrained network for the image stream, such as Carion'20. Alternatively, an unsupervised pretrained network can be used, e.g. Chen'20. These baselines would significantly strengthen the claims.
> > >
> > > Carion'20, End-to-End Object Detection with Transformers.
> > > Chen'20, Exploring Simple Siamese Representation Learning.

---

> > > > ### Author Response · Authors · 2021-08-25
> > > > **Re: Reviewer's response**
> > > >
> > > > Thank you for the response. We are glad the reviewer appreciates the added baselines and clarifications that were requested in the original response.
> > > >
> > > >
> > > > ## Comment on the ‘generality’ of the approach
> > > >
> > > > Please see the general response above regarding the generality of our approach. We have demonstrated our approach on 10 simulated tasks (each with 1000s of unique instances) and 9 real-world tasks, all with diverse language expressions, particularly in the context of language-grounding for robotics. We have also added a section dedicated (Appendix H) to discuss the limitations of our approach.
> > > >
> > > >
> > > > ## Suggestion for another baseline with pre-trained object-detectors
> > > >
> > > > We thank the reviewer for this suggestion. We can run another RN50-BERT baseline with SimSiam’s and DETR’s pretrained weights. Although, we aren't sure if implementation, training, evaluation, and writeup can be completed on short-notice within the Aug 31 deadline.
> > > >
> > > > There are many existing detectors with ResNet-50 backbones that were trained on vision tasks (classification, detection, etc.) We provide a comparison to one such baseline with our ImageNet pretrained RN50 with BERT model. However, our point is that none of these image pre-trained ResNets (e.g. DETR, SimSiam etc.) consider **aligned vision and language** representations. Since CLIPort is focused on language+manipulation, our choice of CLIP-ResNet50 is specifically motivated by CLIP’s multimodal representations for language+vision, which has achieved state-of-the-art results in vision and language tasks (Radford et. al, 2021; Shen et. al, 2021). In contrast, object-detectors are restricted to representations from vision->class models that are **not aligned** to language-only encoders like BERT.
> > > >
> > > > We, the authors, are particularly excited by this new paradigm enabled by CLIP that visual representations don’t have to be restricted to a specific set of classes like with traditional object/instance detectors. There are no existing datasets or detectors that work for an arbitrary number of objects like cherries, pliers, chess boards etc. except for CLIP which was trained with millions of image-caption pairs from the internet.

---

### Official Review · Reviewer_Hq8X · 2021-07-19

**Originality:** Good
**Technical Quality:** Very Good
**Clarity Of Presentation:** Very Good
**Impact:** 3

**Recommendation:**

Weak Reject: I recommend rejecting the paper, but will not argue for my recommendation if the majority of other reviewers have a different opinion.

**Summary:**

The paper presents a framework for learning language-conditioned, vision-based manipulation tasks. The high-level idea is to separate the spatial aspect of manipulation (e.g., where to pick and place) from the language conditions that specify the manipulation target. The goal is to generalize to new language commands by leveraging a pretrained vision-language model. The visual manipulation framework builds on the Transporter network (Zeng et al., 2020), which models multistep manipulation as a dense image prediction problem. The language condition is supplied through a pretrained CLIP vision-language embedding model (Radford et al., 2021). By conditioning the Transporter prediction on the CLIP embedding and training on a fixed set of demonstrations, the framework is shown to zero-shot generalize to follow new language commands (e.g., picking blocks of unseen colors). The method is evaluated on a set of simulated and real-world tabletop manipulation tasks.

**Issues:**

See "weaknesses"

**Reviewer Expertise:**

Very good: Comprehensive knowledge of the area

**Strengths And Weaknesses:**

I like the overall premise of the paper: large-scale vision-language models should be able to help visual-manipulation models to generalize better. I also like the overall execution of the paper: the proposed method is a simple extension of the Transporter model to incorporate language conditions and is evaluated extensively on both simulated and real-world manipulation tasks. However, a few doubts remain on the main thesis: aligning vision-language representation with visual-manipulation representations brings “the best of both worlds”.

First of all, to what extent does the “spatial / geometric” aspect of manipulation benefit from the “semantic” aspect of manipulation? Intuitively, CLIP is an excellent medium for generalizing to unseen visual concepts and combination of visual concepts in *visual recognition tasks*. However, I struggle to understand what kind of synergies between vision-language embedding and manipulation can be exploited beyond, e.g., correctly recognizing a new object that’s not shown in the demonstration because the pretrained CLIP model has that object before.

For example, if the Transporter model is never trained with pick-and-place an awkwardly-shaped or a slippery mug, correctly recognizing that it is a mug shouldn’t help with its manipulation. In other words, if pick-place tasks are mostly about understanding the geometric/dynamic aspect of the objects, why would a vision-language model help with its generalization? We may further extrapolate this argument: if we have a detection model that can correctly localize novel objects (e.g., 2D position & orientations) given language commands, to what extent do the language-conditioned Transporter network benefit from the CLIP embedding?

Second, language commands contain information beyond object categories and attributes. For example, in the example given by the paper “pack all the blue and yellow boxes in the brown box”, to what extent does the relationship “in the” constitute a reusable linguistic concept beyond identifying a task? For example, would the model be able to generalize to “place a dice in the mug'' or “grasp the mug in the cabinet”? I understand that the paper does not claim generalization to novel relationships or non-object-related concepts. But it is important to discuss such limitations or even present negative results.

Minor:
-
Line 165: what does it mean to “downsample” the goal embedding? Are there downsampling operations like strided 1-D convolution involved or is it just a linear projection?

**Summary Of Recommendation:**

I am overall neutral about the paper. I'm recommending weak reject right now but it's more like a "borderline". I will likely change my rating after seeing additional discussion / caveats about the main thesis of the paper and further experiments regarding the limitations.

---

> ### Author Response · Authors · 2021-08-23
> **Response to Reviewer Hq8X - Part 1**
>
> We thank R-Hq8X for raising these two points (copied below). These are great questions for understanding the limitations of CLIPort. Accordingly, we have added Appendix Section H to discuss the limitations of our framework. While both these limitations are beyond the scope of our problem setup, we believe, in its current form, CLIPort already provides a new set of capabilities for tabletop manipulation that simply didn’t exist before. Please see the general response above where we reiterate our claims and contributions. But nonetheless we discuss these two limitations with illustrative examples below.
>
> _“First of all, to what extent does the “spatial / geometric” aspect of manipulation benefit from the “semantic” aspect of manipulation? ... I struggle to understand what kind of synergies between vision-language embedding and manipulation can be exploited beyond, e.g., correctly recognizing a new object that’s not shown in the demonstration because the pretrained CLIP model has that object before.”_
>
> _“For example, if the Transporter model is never trained with pick-and-place an awkwardly-shaped or a slippery mug, correctly recognizing that it is a mug shouldn’t help with its manipulation. In other words, if pick-place tasks are mostly about understanding the geometric/dynamic aspect of the objects, why would a vision-language model help with its generalization?_
>
> Firstly, we do not claim that CLIP’s representations are directly applicable for zero-shot grasping of novel object types. In other words, we are not competing with grasping approaches like DexNet (Mahler et. al, 2017) or GraspNet (Mousavian et. al, 2019) etc. CLIP is a pure vision-language model and has no understanding of physical affordances, actions or any other physical property. But in CLIPort, we fine-tune CLIP’s existing visual representations to produce visual affordance predictions. For example, in “put the pliers in the brown box”, CLIP can _recognize_ ‘pliers’ but it does not know anything about how to manipulate pliers. Through demonstrations we train CLIPort to grasp the pliers by the handle, and maybe at a specific point on the handle that is conducive to the task specified in the language. This learnt affordance can then be reused in the context of a new task where the pliers need to be placed in a different location or used to achieve a different goal. The compositional nature of language allows us to compose novel goals with these learnt affordances, and this is what we evaluate in our large-scale experiments in the Results section. To summarize, our framework is always used in a one-shot, few-shot, or many-shot setting specifically for language-conditioned manipulation, and not in a zero-shot grasping setting.
>
> Our main motivation for using CLIP in particular is that it provides a powerful visuo-linguistic prior for reusing these language-conditioned affordances. To illustrate this, we have added Fig 10 in the appendix ([see this link](https://i.imgur.com/odNMsZE.jpeg)) to showcase a qualitative example of one-shot learning. Here the task is again to “put the pliers in the brown box”. The CLIPort (multi) model has only seen one training example of pliers – _literally_ just one image-action pair with pliers and the word ‘pliers’ in the entire training set. As described by R-Hq8X and AC-uaBQ, we test the model on 3 unseen instances of pliers of different colors and shapes. Note that despite the distractor objects and pose variation, the model is still able to grasp the handles correctly for 2/3 pliers. The third plier, which causes a failure, is substantially different in shape to the training plier and is out-of-distribution for the model to effectively generalize. Despite the grasping failure, it’s still able to localize the plier correctly among the distractor objects, and with a few more training examples it might be able to improve grasping performance. The only reason this one-shot pliers example is feasible is because CLIP has a strong visual understanding of objects and language expressions that tabula rasa frameworks like Transporter-only lack.
>
> Secondly, the main generalization results in the paper are over goals and visual attributes. For example, does having seen pink blocks in “put the pink block in the brown bowl” help in solving “pack all the pink and yellow blocks in the brown box” having never seen pink blocks or the word ‘pink’ in the context of the packing task. Here we use color as an example, but this transfer is applicable to any visual attribute that an internet-pretrained visual model would understand like texts, textures, shapes and categorical names. Again, we do not claim zero-shot generalization to grasping novel objects. Any zero-shot generalization is mostly restricted to novel goals and unseen visual attributes. And we believe this is a very useful skill for conditioning manipulation policies.

---

> > ### Author Response · Authors · 2021-08-23
> > **Response to Reviewer Hq8X - Part 2**
> >
> > _We may further extrapolate this argument: if we have a detection model that can correctly localize novel objects (e.g., 2D position & orientations) given language commands, to what extent do the language-conditioned Transporter network benefit from the CLIP embedding?”_
> >
> > “If we have a detection model” – this is a big “if”. This hypothetical detection model would need to detect chess pieces, all square blocks on the chessboard, every pinto bean, all cherry stems, every configuration of the rope, etc. There is no existing detector out there that works out of the box for these tasks, so a custom detector would need to be trained in a supervised manner. This involves tediously annotating segments of chessboard squares, pinto beans etc. where in some cases it might not even be feasible to do so – e.g. the cherry stems that are barely a few pixels. Even if this hypothetical detection model can be trained this doesn’t mean that the manipulation task is solved. You still need a policy that can go from the output of your vision model (e.g. instance segmentation) to executable pick-and-place actions. This is a non-trivial problem in general considering all the diverse tasks that we present. In contrast, our framework provides an end-to-end solution where a user can train the perception and action model jointly end-to-end directly through demonstrations.
> >
> >
> >
> > Regarding “do language-conditioned Transporter network … benefit from CLIP embedding”, we provide various baselines in Table 1 and Appendix Table 4 including ImageNet trained ResNet50 with BERT and untrained semantic stream models. CLIP is substantially better than any traditional visual encoder (Radford et. al, 2021) like ImageNet trained ResNet50 that is typically used in robotics settings. CLIP representations have also been used to achieve SoTA results with vision-language models in a range of domains from image-captioning to vision-language navigation (Shen et. al, 2021). We also note again that the original Transporter does not involve any form of goal-conditioning and involves purely imitating demonstrations. Transporter by itself does not work in a multi-task or multi-goal setting because it isn’t conditioned on anything but a specific demonstration itself.
> >
> > _Second, language commands contain information beyond object categories and attributes. For example, in the example given by the paper “pack all the blue and yellow boxes in the brown box”, to what extent does the relationship “in the” constitute a reusable linguistic concept beyond identifying a task? For example, would the model be able to generalize to “place a dice in the mug'' or “grasp the mug in the cabinet”? I understand that the paper does not claim generalization to novel relationships or non-object-related concepts. But it is important to discuss such limitations or even present negative results._
> >
> > You are correct in that CLIPort’s language-grounding capabilities cannot generalize beyond the demonstrated tasks. All language expressions, including parts like “in the”, are grounded in the tasks and demonstrations shown to the model during training. We have added a section in Appendix H to discuss this in detail. Note that language-grounding in unrestricted environments is a broad and open challenge in embodied AI. This sort of generalization is beyond the scope of this work, and perhaps any robotics framework at the moment.
> >
> > But more specifically, “place a dice in the mug” should work assuming the model has leant to grasp the dice in some task and to place objects in a mug from some other task. The language command should be sufficient to compose the two skills together in a new scene and perhaps even with unseen instances of dice and mugs. This is exactly what we evaluate with CLIPort (multi-attr) results in Table 1. We do not claim generalization to novel relationships, but despite this, we show some strong generalization across attributes and instances that is still useful in real-world settings. Please see Appendix H for an extended discussion on grounding complex relationships.
> >
> > It’s likely that the complexity of the tasks in our benchmark is hard to gauge from figures in the paper. We kindly urge R-Hq8X to examine the following evaluation videos for the packing-box-pairs-seen-colors tasks: [video link](https://drive.google.com/drive/folders/1pQTSTpQHPbkEY1mGwiSxLXIS9lQCFsjf?usp=sharing) (also at the bottom of the website). Each of these five evaluation instances involve several blocks of unseen sizes, of randomized quantity, with randomized colors, and randomized poses. Note that CLIPort solves all these tasks with just RGB-D input. We hope that these videos and more results on the website ([cliport.github.io](https://cliport.github.io)) help better communicate the challenges in our language-conditioned benchmark.
> >
> > Radford et. al, 2021, https://arxiv.org/pdf/2103.00020.pdf
> > Shen et. al, 2021, https://arxiv.org/pdf/2107.06383.pdf

---

> > ### Comment · Reviewer_Hq8X · 2021-08-25
> > **Re: Response to Reviewer Hq8X - Part 1**
> >
> > Thanks for the detailed reply and clarifying the key contribution. The additional result indeed looks impressive and does show the few-shot learning capability. However, none of this is clear in the original text, especially the abstract & introduction, e.g, line 14-15: "Our end to-end framework is capable of solving a variety of language-specified tabletop tasks from packing unseen objects to folding cloths, all without ...". There is no mentioning that the method is targeted at few-shot learning setting until the experiment section. I strongly suggest the authors to revise the paper to reflect the scope of the paper.
> >
> > At the same time, I'm still not sure how it would compare to a detector-based method. It is true that the intermediate representation of the detector (e.g., axis-aligned bbox) prohibits such method from being applied to more tricky manipulation settings out of the box. However, the nature of the demonstration used by CLIPport is very similar to that of a bounding box, except it takes the form of a grasp / place location and orientation. I'm still trying to understand how it fundamentally differs from a detection model that uses CLIP as its backend and instead of outputing box locations and sizes, generating the grasping / placing location and orientation.

---

> > > ### Author Response · Authors · 2021-08-25
> > > **Action-Centric vs. Object-Centric Perception**
> > >
> > > Thank you for the response. We are glad the reviewer appreciates the new results.
> > >
> > > We agree that the few-shot results weren’t emphasized enough in the introduction. We will update the paper to reflect this. Thank you for the suggestion.
> > >
> > > Please see our previously posted [response below](https://openreview.net/forum?id=9uFiX_HRsIL&noteId=7MC7EkK9YY) on training “detection models”.
> > >
> > > We see what you mean here that there is some sort of connection between detecting objects with bounding-boxes and predicting actions with CLIPort. But CLIPort (and Transporter) take a fundamentally different approach to perception that is action-driven (Gibson, 1986). Broadly speaking, we can categorize perception for manipulation into two paradigms: object-centric perception vs. action-centric perception. Object-centric perception is about “detecting objects”, but action-centric perception is about “detecting actions”. Here we illustrate some examples where traditional bounding-box detections of objects would not be feasible:
> > >
> > > 1. [Unfolding Cloth](https://i.ibb.co/FVcp91X/Screen-Shot-2021-08-25-at-10-39-04-AM.png) – There is no object  **to detect** in the placement location
> > >
> > > 2. [Packing Boxes](https://i.ibb.co/2NBKjHW/Screen-Shot-2021-08-25-at-10-39-07-AM.png) – The placements are a multimodal distribution with precise positions that depends on the size and shape of the object
> > >
> > > 3. [Cherry Picking](https://i.ibb.co/ZNb4xBs/Screen-Shot-2021-08-25-at-10-39-10-AM.png) – The cherries have to be picked by the stem in order to avoid destroying the fruit. The stems are barely a few pixels in the image.
> > >
> > > 4. [Sweeping Beans/Blocks](https://i.ibb.co/JvWqs6t/Screen-Shot-2021-08-25-at-10-39-13-AM.png) – We are **not ‘detecting’ every bean**, but we can still sweep every bean into the specified zone.
> > >
> > > We realize that the introduction might have not sufficiently stressed the importance of action-centric vs. object-centric approaches. While there have been a number of works leading up to action-driven perception – like Transporter (Zeng et. al), Form2Fit (Zakka et. al), it might not be familiar to most readers. Accordingly, we will update the paper to make this clearer.
> > >
> > > We also kindly urge the reviewer to take a look at the experiments in the original Transporter paper (Zeng et. al, 2020). In particular, Zeng et al. showed that Transporter outperforms baselines that use object-centric information such as ground truth object poses.
> > >
> > > Gibson, J. J. (1986). The ecological approach to visual perception. Hillsdale, New Jersey: Lawrence Erlbaum Associates.

---

> > > > ### Author Response · Authors · 2021-08-30
> > > > **Any other concerns?**
> > > >
> > > > Please kindly let us know if our response above addresses your concerns.
> > > >
> > > > We have updated the paper (with changes highlighted in blue) to emphasize efficiency of few-shot settings and added a brief discussion on action-centric approaches in the introduction.
> > > >
> > > > We once again thank you for your feedback, and hope you share our excitement about the results presented in the paper: [cliport.github.io](https://cliport.github.io/)

---

### Official Review · Reviewer_WZS2 · 2021-07-24

**Originality:** Excellent
**Technical Quality:** Excellent
**Clarity Of Presentation:** Very Good
**Impact:** 4

**Recommendation:**

Strong Accept: I recommend accepting the paper and will argue for my recommendation even if other reviewers hold a different opinion.

**Summary:**

The authors combine a large pre-trained CLIP vision-and-language model with a Transporter network in order to facilitate training a multi-task policy conditioned on natural language inputs with limited data. They compare against CLIP and Transporter baselines on both trained language-conditioned tasks and transferred languaged-conditioned tasks. They find that this method is able to generalize effectively to novel combinations of attributes (such as seperating the notions of 'blue' and 'cube', so that they can solve tasks involving blue cubes after having seen tasks with blue non-cubes and non-blue cubes). In tasks involving this kind of conceptual decomposition, they dramatically outperform other baselines.

**Issues:**

It's not entirely clear how the authors are moving from the affordance prediction to actions on the actual robot. Since the labelled data all seems to be labelled images rather than action sequences, my best guess would be that the authors are using the affordances to select pre-programmed manipulations to take, and what parameters to call those manipulations with. I'd like to understand this better though, so a little more clarification would be nice.

**Reviewer Expertise:**

Fair: Some knowledge of the area

**Strengths And Weaknesses:**

Strengths:
I find this particular kind of of generalization across attributes extremely interesting, especially considering that they achieve it solely through end-to-end training. Additionally, the demonstration that large pre-trained networks for non-robotics language and vision tasks can be used to enable more efficient training training for robotics seems significant, as data and pre-trained models for these domains are much more accessible. I was also pleasantly suprised at the amount of training time used to achieve this level of result -- 6 GPU days is a decent amount but feasible for most labs.

Weaknesses:
It's not entirely clear how the authors get from the affordance prediction to the actual actions on the robot.

**Summary Of Recommendation:**

This result is both immediately useful and offers up new and promising opportunities for future research. The use of natural language is both incredibly convenient for use and adding new tasks, and suprisingly data-efficient. The work additionally offers interesting new avenues for how large pre-trained networks from other domains could be used to enable more efficient robot learning.

---

> ### Author Response · Authors · 2021-08-23
> **Response to Reviewer WZS2**
>
> _“It's not entirely clear how the authors get from the affordance prediction to the actual actions on the robot.”_
>
> _“It's not entirely clear how the authors are moving from the affordance prediction to actions on the actual robot. Since the labelled data all seems to be labelled images rather than action sequences, my best guess would be that the authors are using the affordances to select pre-programmed manipulations to take, and what parameters to call those manipulations with. I'd like to understand this better though, so a little more clarification would be nice.”_
>
> We thank R-WZS2 for noting this. This is explained in Appendix Section C L657, but there were some details missing in the original submission. Accordingly, we have updated Section C with additional information on policy execution.
>
> All actions (including for tasks like packing, sweeping etc.) are executed through a simple pick and place primitive. The output of CLIPort is (u, v, theta) for Pick and (u’, v’, theta’) for Place, which are argmax values from affordance predictions shown in Figure 4. We take a 32x32 crop of the depth-pointcloud around these points to compute two 3D-centroids for pick and place, respectively. With a known hand-eye calibration, these 3D centroids are used with a motion-planner (RRT*) to move the end-effector in a predefined up-down-up fashion to reach the 3D location, open/close the gripper, and then lift-up. Note that this is a technique frequently used in grasping settings such as DenseObjectNets (Florence et. al, 2018) and DexNet (Mahler et. al, 2017). Only the sweeping and folding controllers are slightly different in that the end-effector doesn’t raise-up from the pick position before moving to the place location. But again, both sweeping and folding actions executed through the same pick and place primitives predicted by the model. While this action-space is limited, it still allows for a wide range of useful tabletop tasks like packing and sorting, particularly in industrial settings. For all sequential tasks involving multiple steps, the model is purely reacting from RGB-D input to RGB-D input after executing each pick and place action without maintaining a history or belief across timesteps.

---

### Author Response · Authors · 2021-09-01
**Copy of Author's initial response to Meta-Review**

Re-posting the original author's response to the meta-reviewer. Original comment date (8/23/2021)


----------------------
We thank the reviewers for their insightful feedback. We are glad that all reviewers appreciate the novelty of using large-scale vision-language models for learning generalizable conditioning for manipulation policies. In particular R-WZS2 highlights the data-efficiency of our end-to-end approach along with the ability to “generalize [to] novel combinations of attributes”. R-HUJp highlights the use of a pre-trained CLIP model as “a promising direction for robotic control”. R-Hq8X notes the simplicity of the solution and the extensive evaluations conducted in both simulated and real-world settings.


# Clarifying Problem Statement and Contributions:

We believe that there is some confusion, particularly with R-Hq8X, regarding our claims, problem scope, and our main contributions. These may not have been sufficiently clear in the paper so we briefly clarify these points before providing detailed responses to the comments from the reviewers and the meta-reviewer.


## Problem Statement:

The problem we consider is that of _language-conditioned learning for precise robotic manipulation from demonstrations_ (see [cliport.github.io](http://cliport.github.io) for examples). Namely, given a small number of language-conditioned manipulation demonstrations we would like our robot to accomplish a manipulation task specified through a language command. This problem statement goes beyond the capabilities of Transporter (Zeng et. al, 2021), which only tackles learning from demonstration with no conditioning mechanism, or Goal-Transporter (Seita et. al, 2021) which relies on goal-images. We chose to focus on language as our conditioning modality since it is a general and flexible way to communicate goals. However, we want to explicitly clarify the things that we are **not attempting to solve**:



* Grasping novel object types for which no relevant grasping demonstrations have been given.
* Following arbitrary (out of distribution) language instructions, e.g. “clean the kitchen”, for which no demonstrations have been given.

Although we would eventually like to achieve such capabilities, this work should be considered as a first step in that direction. We would like to point out that the only supervision that our system receives comes from the demonstrations, and thus we don’t attempt or claim to be able to generalize outside of this distribution. An interesting direction for future work would involve collecting a large number of diverse demonstrations, similar to how CLIP is trained with a diverse set of image-caption pairs, to investigate whether this enables zero shot generalization to novel objects and tasks. However, this is beyond the scope of the paper.


## Reiterating Contributions

Here we reiterate the capabilities of our framework to emphasize that (to the best our knowledge) there is no other existing end-to-end system that is comparable to CLIPort:



* Language-conditioned policies for precise manipulation including sweeping particles, manipulating ropes, packing objects, and moving chess pieces.
* An end-to-end framework without any object models, poses, segmentations, bounding boxes, language parser, or history. Particularly in the context of language-grounding for spatial manipulation, this hasn’t been done before.
* A single, multi-task model for sweeping, manipulating ropes, packing boxes etc.
* Works on a real-robot!
* All 9 real-world tasks trained with just 179 image-action pairs.

We kindly urge the reviewers and the meta-reviewer to examine qualitative results on our project website: [cliport.github.io](http://cliport.github.io). The results at the bottom also showcase the complexity of the simulated tasks in both single-task and multi-task settings. In its current state, we believe CLIPort already provides new capabilities for a wide range of industrial and assistive robotics tasks that were simply not possible before. With these clarifications and qualitative results, we hope reviewers will share our excitement regarding the results presented in the paper.

---

### Author Response · Authors · 2021-09-01
**Copy: Summary of updates to the paper**

Re-posting as this became hidden when meta-review was made private.

# General Updates

We have updated the paper to address all the reviewers’ comments. A brief summary of these changes:



* RN50-BERT baselines for all tasks in Table 1 and Figure 3.
* No pre-training baselines in Appendix Table 4.
* Added Appendix H – a dedicated section that discusses the limitations of our approach.
* Added Figure 10 – a qualitative example to address R-Hq8X’s concern on the usefulness of CLIP for language-conditioned grasping.
* More details on policy execution in Appendix Section C.
* 3 new real-world tasks and results: cherry picking, rope manipulation, and reading text.
* More related work.
* Project website with qualitative results: [cliport.github.io](http://cliport.github.io)
* Attached the appendix to the main PDF for easy-reading.

---

### Author Response · Authors · 2021-10-08
**Camera-Ready Updates**

We thank the reviewers and AC for their insightful feedback. The discussions have been very helpful in improving the paper.

Some updates for the camera-ready version:
- Following standard practice, we have added a separate validation and test set to avoid over-optimizing on evaluation tasks in future works. The test results are in the Results section, and the validation results are in the Appendix. All references in the prose  to quantitative numbers have been updated accordingly.
- Added more related work.
- Minor edits to the introduction.
- Fixed an error in the Full Architecture diagram in the appendix.
- The benchmark, code, pre-trained models, and videos are available at: [cliport.github.io](https://cliport.github.io/)

We look forward to exciting future works at the intersection of Robotics, NLP, Vision, and ML.

---

### Meta-Review · Area_Chair_uaBQ · 2021-08-10

**Recommendation:** Accept (Poster)
**Confidence:** 5

**Metareview:**

CLIPort combines CLIP (a joint language and image embedding) with TransporterNets (a method for performing data efficient BC on sparse action manipulation problems) to enable multi-task NLP conditioned policy learning. Multiple reviewers praised the techniques ability to generalize via CLIP pretraining and it’s leveraging of large-scale internet datasets, as well as the perceived data efficiency of the approach. The authors addressed the bulk of reviewer feedback, adding clarifications for limitations as well as additional real-world experiments and baselines. The paper achieves compelling real-world results and should be of interest to the CoRL community.

---

> ### Author Response · Authors · 2021-08-23
> **General Response**
>
> We thank the reviewers for their insightful feedback. We are glad that all reviewers appreciate the novelty of using large-scale vision-language models for learning generalizable conditioning for manipulation policies. In particular R-WZS2 highlights the data-efficiency of our end-to-end approach along with the ability to “generalize [to] novel combinations of attributes”. R-HUJp highlights the use of a pre-trained CLIP model as “a promising direction for robotic control”. R-Hq8X notes the simplicity of the solution and the extensive evaluations conducted in both simulated and real-world settings.
>
>
> # Clarifying Problem Statement and Contributions
>
> We believe that there is some confusion, particularly with R-Hq8X, regarding our claims, problem scope, and our main contributions. These may not have been sufficiently clear in the paper so we briefly clarify these points before providing detailed responses to the comments from the reviewers and the meta-reviewer.
>
>
> ## Problem Statement:
>
> The problem we consider is that of _language-conditioned learning for precise robotic manipulation from demonstrations_ (see [cliport.github.io](http://cliport.github.io) for examples). Namely, given a small number of language-conditioned manipulation demonstrations we would like our robot to accomplish a manipulation task specified through a language command. This problem statement goes beyond the capabilities of Transporter (Zeng et. al, 2021), which only tackles learning from demonstration with no conditioning mechanism, or Goal-Transporter (Seita et. al, 2021) which relies on goal-images. We chose to focus on language as our conditioning modality since it is a general and flexible way to communicate goals. However, we want to explicitly clarify the things that we are **not attempting to solve**:
>
>
>
> * Grasping novel object types for which no relevant grasping demonstrations have been given.
> * Following arbitrary (out of distribution) language instructions, e.g. “clean the kitchen”, for which no demonstrations have been given.
>
> Although we would eventually like to achieve such capabilities, this work should be considered as a first step in that direction. We would like to point out that the only supervision that our system receives comes from the demonstrations, and thus we don’t attempt or claim to be able to generalize outside of this distribution. An interesting direction for future work would involve collecting a large number of diverse demonstrations, similar to how CLIP is trained with a diverse set of image-caption pairs, to investigate whether this enables zero shot generalization to novel objects and tasks. However, this is beyond the scope of the paper.
>
>
> ## Reiterating Contributions
>
> Here we reiterate the capabilities of our framework to emphasize that (to the best our knowledge) there is no other existing end-to-end system that is comparable to CLIPort:
>
>
>
> * Language-conditioned policies for precise manipulation including sweeping particles, manipulating ropes, packing objects, and moving chess pieces.
> * An end-to-end framework without any object models, poses, segmentations, bounding boxes, language parser, or history. Particularly in the context of language-grounding for spatial manipulation, this hasn’t been done before.
> * A single, multi-task model for sweeping, manipulating ropes, packing boxes etc.
> * Works on a real-robot!
> * All 9 real-world tasks trained with just 179 image-action pairs.
>
> We kindly urge the reviewers and the meta-reviewer to examine qualitative results on our project website: [cliport.github.io](http://cliport.github.io). The results at the bottom also showcase the complexity of the simulated tasks in both single-task and multi-task settings. In its current state, we believe CLIPort already provides new capabilities for a wide range of industrial and assistive robotics tasks that were simply not possible before. With these clarifications and qualitative results, we hope reviewers will share our excitement regarding the results presented in the paper.
>
>
> # General Updates
>
> We have updated the paper to address all the reviewers’ comments. A brief summary of these changes:
>
>
>
> * RN50-BERT baselines for all tasks in Table 1 and Figure 3.
> * No pre-training baselines in Appendix Table 4.
> * Added Appendix H – a dedicated section that discusses the limitations of our approach.
> * Added Figure 10 – a qualitative example to address R-Hq8X’s concern on the usefulness of CLIP for language-conditioned grasping.
> * More details on policy execution in Appendix Section C.
> * 3 new real-world tasks and results: cherry picking, rope manipulation, and reading text.
> * More related work.
> * Project website with qualitative results: [cliport.github.io](http://cliport.github.io)
> * Attached the appendix to the main PDF for easy-reading.

---

### Decision · Program_Chairs · 2021-09-13

**Decision:**

Accept (Poster)

**Comment:**

CLIPort combines CLIP (a joint language and image embedding) with TransporterNets (a method for performing data efficient BC on sparse action manipulation problems) to enable multi-task NLP conditioned policy learning. Multiple reviewers praised the techniques ability to generalize via CLIP pretraining and it’s leveraging of large-scale internet datasets, as well as the perceived data efficiency of the approach. The authors addressed the bulk of reviewer feedback, adding clarifications for limitations as well as additional real-world experiments and baselines. The paper achieves compelling real-world results and should be of interest to the CoRL community.